# Fourth Ventricle Epidermoid Cyst: Case Report of Precision Telovelar Microsurgery, Functional Preservation, and Lifelong Surveillance

**DOI:** 10.3390/diagnostics15202600

**Published:** 2025-10-15

**Authors:** Daniel Costea, Nicolaie Dobrin, Catalina-Ioana Tataru, Corneliu Toader, Răzvan-Adrian Covache-Busuioc, Matei Șerban, Octavian Munteanu, Ionut Bogdan Diaconescu

**Affiliations:** 1Department of Neurosurgery, “Victor Babes” University of Medicine and Pharmacy, 300041 Timisoara, Romania; costea.damiel@umft.ro; 2”Prof. Dr. Nicolae Oblu” Emergency Clinical Hospital, 700309 Iasi, Romania; 3Clinical Department of Ophthalmology, “Carol Davila” University of Medicine and Pharmacy, 020021 Bucharest, Romania; 4Department of Ophthalmology, Clinical Hospital for Ophthalmological Emergencies, 010464 Bucharest, Romania; 5Puls Med Association, 051885 Bucharest, Romania; 6Department of Neurosurgery, “Carol Davila” University of Medicine and Pharmacy, 050474 Bucharest, Romania; 7Department of Vascular Neurosurgery, National Institute of Neurology and Neurovascular Diseases, 077160 Bucharest, Romania; 8Discipline of Anatomy Department II of Morphological Sciences, “Carol Davila” University of Medicine and Pharmacy, 050098 Bucharest, Romania

**Keywords:** epidermoid cyst, fourth ventricle tumor, posterior fossa surgery, telovelar approach, cerebellar ataxia, brainstem compression, cranial nerve preservation, diffusion-weighted MRI, microsurgical resection, long-term outcomes

## Abstract

**Background and Clinical Significance**: Fourth ventricular epidermoid cysts are among the least frequently encountered intracranial tumors (less than 1%). Their slow growth pattern along cisternal and subarachnoid spaces, and their close proximity to neurovascular structures (brainstem–cerebellar), create difficulty for surgical treatment. Total removal is often complicated by the capsule’s adherence to eloquent structures and requires a thoughtful surgical approach of weighing radical resection versus neurologic/function preservation. This case description provides an example of using careful clinical–radiological correlation and anatomy-dissecting microsurgery as a method of permanent decompression and neurologic recovery with low operative risk. **Case Presentation**: A 57-year-old female presented with impaired stability of gait, gaze-evoked nystagmus, appendicular ataxia, minimal ipsilateral hypotonia, and mild bulbar dyscoordination. Imaging (MRI, MRA) revealed a large, lobulated mass that was lobulated and avascular centered in the left cerebellar hemisphere, with an extension into the vermis and cisterna magna, and partial filling of the fourth ventricle with classic epidermoid imaging. Resection was performed via a midline suboccipital telovelar approach with microsurgery, relying on native arachnoid planes and quadrant opportunities of decompression, while preserving critical neurovascular structures. A thin rim of capsule intimately adherent to the floor of the ventricle was intentionally left to minimize irreversible cranial nerve injury. Histology showed keratinizing stratified squamous epithelium with laminated keratin and cholesterol clefts. Following resection, truncal stability, limb coordination, and ocular pursuit improved without additional deficits. Initial and 3-month postoperative MRI showed total decompression, re-established CSF pathways, and no recurrence. **Conclusions**: This case demonstrates that maximal safe resection (with function preservation) through natural anatomy corridors can achieve excellent neurologic results in fourth ventricular epidermoids. Lifelong MRI surveillance will be needed due to the srisk of delayed recurrence even after near-total resection.

## 1. Introduction

Intracranial epidermoid cysts are rare, benign, and congenital lesions that develop from ectodermal cell inclusions trapped during the incomplete closure of the neural tube between the third and fifth weeks of embryonic development [1]. These inclusions possess a lifelong capacity for keratin production, gradually producing concentric layers of keratinized debris and cholesterol crystals, surrounded by a thin lining of stratified squamous epithelium. With slow desquamation over the decades, epidermoid cysts result in a space-occupying lesion which expands along natural cisternal and subarachnoid routes without true parenchymal invasion, molding to surrounding neurovascular structures [2].

Epidermoid cysts account for 0.2% to 1.8% of all primary intracranial tumors described globally, with some large series based on national registries showing prevalence at the lower end of the range in countries that provide access to MRI imaging studies. The incidence of epidermoid cysts appears to be similar worldwide, although overall rates may vary regionally, likely reflecting how diagnostic resources impact reporting rates and the ability to capture surgical cases [3]. Most series suggest that the cerebellopontine angle is the most common site (40% to 50% of epidermoid cysts), followed by the parasellar region (10% to 15%) and other cisternal compartments. Location in the fourth ventricle is particularly uncommon, at 5% to 18% of all intracranial epidermoids, and extraordinarily rare in children, with this dearth reflected in the literature. Most of the modern understanding of epidermoids of the fourth ventricle is derived from a small, single-institution series of patients or isolated case reports [4].

The mean age at which an epidermoid cyst is diagnosed most often occurs in the third to fifth decades of life, corresponding to a prolonged latent phase during which this slow expansion gradually impairs eloquent neuroanatomy. Although a few reports have noted a slight male predominance, the most recent pooled analyses do not show a statistically significant sex bias [5]. Clinical presentation is closely related to the spatial relationships of the lesion: midline growth within the fourth ventricle usually compresses the cerebellar vermis, resulting in truncal ataxia and gait instability; lateral growth into the hemispheres can cause ipsilateral limb dysmetria and dysdiadochokinesia; compression of the cerebellar peduncles can interfere with gaze-holding, ocular tracking, and vestibular integration; contact with the medulla may more subtly affect bulbar coordination. Obstruction of the fourth ventricular outlets, especially the foramen of Magendie, can cause progressive obstructive hydrocephalus that can present as headache, nausea, vomiting, and papilledema [6,7].

From a radiologic perspective, fourth-ventricular epidermoid cysts have a defined imaging profile. While hypodense on non-contrast CT, these lesions classically conform to the shape of the ventricle and adjacent cisterns. The gold-standard modality is MRI: typically iso- to hypointense to CSF on T1-weighted sequences and hyperintense on T2-weighted sequences, with “dirty FLAIR” suppression and contrast associated with the incomplete suppression on FLAIR (showing diffusion restriction on DWI, which is important for differential diagnosis), and little or no solid enhancement on post-contrast imaging, allowing it to be distinguished from cystic medulloblastomas, ependymomas, dermoid cysts, and rare cystic metastases [8].

Histopathologically, the cysts are defined by a thin, avascular capsule of keratinizing stratified squamous epithelium that sits on the laminated core of anucleate keratin. Cholesterol clefts, inflammatory macrophages, and occasional calcifications may be seen. There is no adnexal skin component, which distinguishes them from dermoid cysts. Molecular profiling is still in its infancy, with one or two studies starting to report some scoring of keratin expression and epithelial proliferation indices that may be of significance for prognostic purposes [2].

From a surgical standpoint, managing fourth-ventricular epidermoid cysts is technically demanding. The tumor capsule can often become deeply intermingled along the ependymal surface of the ventricular floor, the root entry zones of cranial nerves IX–XII, and small perforating vessels and branches—a very common example being from the posterior inferior cerebellar artery (PICA) [9]. To deal with this, and without injury to functionally important neural structures, one generally has a strategy of extra-axial dissection along the preserved arachnoid planes with maximal safe resection and minimal or no injury to neural structures at risk. Historically, a surgical trajectory leaning toward curative radical capsule removal was commonplace, but as modern surgical principles continue to evolve, an increased emphasis on functional preservation over aggressive pursuit of microscopic remnants when critical structures are involved has seen a gradual emergence [10]. The telovelar approach—access via the natural meninges fissure without splitting the vermis—has rendered previous midline transvermian access of antiquity, reducing both postoperative cerebellar mutism and truncal ataxia. A variety of intraoperative adjuncts, such as neuronavigation or neuroendoscopy, may sometimes be employed, but many authors emphasize that anatomical familiarity will remain the main protection for the surgeon [11].

Overall, follow-up should be good when near-total or total excision is achieved. Although not negligible, complete excision has low recurrence rates, and case(s) of regrowth well over ten years later have been described, emphasizing that long-term radiological monitoring is essential. The most commonly published functional outcome measures include the Karnofsky Performance Status (KPS) and modified Rankin Scale (mRS) to describe performance in the postoperative period, with morbidity further increased by the risk of neurological complications involving lower cranial nerve deficits or brainstem injury, if the surgeon forcibly alters the adherence characteristics of the capsule [12].

Although the absolute number of patients affected is small, medically, the socioeconomic consequences could be considerable in cases of delayed diagnosis where hydrocephalus, brainstem compression, or protracted rehabilitation is involved. Managing patients surgically in potentially anatomically complex posterior fossa locations is a risky undertaking, and any identified neurological deficit postoperatively, even transiently, is a potential hazard to the patient’s quality of life and healthcare resource utilization [13].

In a similar vein, well-documented cases—whether numbers are few or not—carry disproportionate value. The intent of the current report is to support a modern understanding of fourth-ventricular epidermoid cysts by providing a comprehensive account of clinical unfolding, radiological–anatomical correlations, operative approach, and postoperative progress, followed by sequential imaging. It is the case that authors desire to combine modern surgical principles with patient-specific anatomical realities, illuminating site-dependent challenges in balancing clearance of lesion and preserving as much function as possible. In anchoring the case report within the existing—albeit not comprehensive—literature, it is hoped that it will help future clinicians navigate the management of these comparatively rare but clinically relevant pathologies.

## 2. Case Presentation

A 57-year-old right-handed woman was referred to us from a secondary care site for neurosurgical assessment after several weeks of progressive neurological deterioration characterized by worsening gait and postural instability, with some requirement for assistance to move around for even basic mobility.

On admission, she was alert, attentive, and well-oriented. Speech was fluent but dysrhythmic with irregular timing of syllables, fragmented prosody, vowel prolongations, and soft consonants consistent with cerebellar dysarthria due to impaired temporal coordination of speech motor output. Ocular assessment revealed coarse, conjugate horizontal nystagmus on sustained gaze left, with a slow drift toward primary position followed by corrective saccadic movement, and a similar but milder issue occurring on sustained gaze right. Smooth pursuits were intermittently interrupted by small corrective saccades, and after more than 3 s of eccentric gaze, she noted horizontal diplopia resolving upon returning to midline, indicative of fatigable gaze hanging under cerebellar control. Facial motor functions were symmetric at rest and to volitional movement. Rapid alternating movements of both lips and tongue had subtle but consistent hesitation, indicative of impaired synchronization of bulbar muscles. Palate elevation was symmetric, the gag reflex was intact, and swallowing was safe, with some focal asynchronous contraction as it appeared slightly arrhythmic.

Motor exam showed left-sided ataxic hemiparesis, reduced strength (MRC 4/5) of both arms and legs, but the greatest deficit was in coordination. The finger-to-nose test demonstrated consistent overshoot, coupled with variable damped oscillations and an intention tremor that increased with fatigue. The heel-to-knee-to-shin demonstrated laterally directed deviation with a non-smooth, erratic trajectory. Rapid alternating movements of her left forearm and the left foot quickly became arrhythmic, a classic dysdiadochokinesia finding. Passive movement showed very soft hypotonia without spasticity present. Deep tendon reflexes were symmetric, with a slight decrease on the left side. No clonus was noted, and plantar responses were bilaterally flexor. Axial stability was also severely impaired. When seated, she demonstrated multidirectional, unpredictable sway—ranging from slow drift to rapid repositioning. This increased when her eyes were closed. When able to stand unsupported, she maintained her posture for less than four seconds without external support. The instability was universally arrhythmic, with diffuse, inconsistent rather than coherent vectoring, which matched the severe dysfunction of the vermis and disorganization of vestibulospinal integration. All modalities—light touch, pin prick, temperature, vibration, and joint position—were intact over all four extremities without neglect, extinction, or other sensory aberration. In summary, the extinction pattern of appendicular ataxia, truncal instability, gaze-evoked nystagmus, subtle bulbar discoordination, and mild hypotonia was suggestive of a left cerebellar hemispheric lesion extending midline, where it compressed the fourth ventricle and had mass effect upon the brainstem. The gradual course and lack of systemic signs indicated a slowly expanding extra-axial lesion, with potential for acute neurological decline caused by obstructive hydrocephalus. Assessment of Functional Status: Prior to surgery, the KPS score was 60, indicating severe functional limitations and dependence on daily living skills. The mRS score was three, indicating the patient was moderately disabled but required some supervision. Gait and balance were severely impaired prior to surgery, with a Scale for the Assessment and Rating of Ataxia (SARA) score of 34 out of 56, indicating marked cerebellar dysfunction with severe truncal and appendicular ataxia. The SARA scale was chosen for its reproducibility, ease of bedside use, and sensitivity to changes in gait, stance, and limb coordination in cerebellar disease. Diagnostic Reasoning/Differential Diagnosis: Progressive ataxia, gaze-evoked nystagmus, subtle bulbar signs, and narrowing of the fourth ventricle would systematically localize the lesion to the left cerebellar hemisphere and midline. The above arguments, combined with the lack of supporting systemic process or parenchymal invasion, led to a differential diagnosis primarily of extra-axial lesions, such as epidermoid cyst, arachnoid cyst, or meningioma. The absence of enhancement on contrast MRI, and a growth pattern that insinuated those of an epidermoid cyst vs. a meningioma; prominent diffusion restriction and internal heterogeneity were sufficient to make it different from an arachnoid cyst. Hemangioblastoma or metastasis was excluded due to the lack of any vascular blush or feeding vessels according to MRA.

High-resolution MRI with selective MRA was obtained immediately after the examination to determine the spatial extent of the mass, vascular relationships to it, and to evaluate the current status of cerebrospinal fluid dynamics. Sagittal MRA reconstruction (Figure 1) revealed a large extra-axial mass in the left posterior fossa that centered on the cerebellar hemisphere and extended medially into the vermis. Posterior fossa arteries were lengthened and changed their course: the posterior inferior cerebellar artery (PICA) appeared to follow an elongated superior–posterior course along the dorsal surface of the mass with maintained continuity and caliber, the anterior inferior cerebellar artery (AICA), and small cerebellar perforators were also pushed to the left but appeared to be unaffected. The gradual vascular displacement without distortion, alteration, or irregularity would be consistent with chronic adaptive remodeling of the vessels; acute distortion did not occur, and there was no vascular blush, arteriovenous shunt, and/or feeding-vessel hypertrophy, identifying the extra-axial mass as an avascular, space-occupying lesion consistent with an epidermoid tumor.

Axial T2-weighted sequences (Figure 2A,B) revealed a large, lobulated mass occupying most of the left cerebellar hemisphere, insinuating between folia and extending across the midline into the inferior vermis. The lesion was markedly hyperintense compared to cerebellar parenchyma, with internal heterogeneity suggesting layered keratinaceous content. The inferior pole of the mass descended into the cisterna magna, and its medial aspect encroached on the fourth ventricle, partially effacing it and displacing its floor anteriorly over the dorsal medulla. This displacement explained the patient’s severe truncal instability, as vermian compression interrupts integration of proprioceptive and vestibular input, and also her left-sided appendicular ataxia, arising from disruption of cerebellar hemispheric coordination pathways projecting through the dentate nucleus and superior cerebellar peduncle.

Susceptibility-weighted imaging (Figure 2C) showed tiny punctate hypointense lesions foci within lesions, suggestive of microcalcifications or hemosiderin secondary to chronic contact with the tentorium, vasculature, and pia, all indicating a long-standing, slowly growing mass. Post-contrast T1-weighted imaging (Figure 2D) displayed no enhancement, considering both lesions were different from a meningioma, hemangioblastoma, or metastasis. Coronal T2-weighted sequences (Figure 2E) demonstrated inferior and lateral extension to the foramina of Luschka, stenosing the lateral recesses of the fourth ventricle and displacing the cerebellar peduncles, most notably left. This accounted for the gaze-evoked nystagmus and pursuit deficits, considering vestibulocerebellar input and flocculonodular output. Cerebellomedullary junctional contact was responsible for the subtle bulbar discoordination secondary to pressure to the dorsal medulla near the nucleus ambiguus. Sagittal FLAIR (Figure 2F) showed an anteriorly displaced brainstem; however, the dorsal medulla was compressed against the wall of the fourth ventricle, with the pontomedullary junction displaced anteriorly. The prepontine cistern was partly effaced, and the basal cisterns were narrowed but still patent; however, inferior edema was enough for acute obstructive hydrocephalus.

Overall, the clinical–imaging cumulative data provided support for the diagnosis of an extra-axial, slow-growing epidermoid cyst, with lobulated margins, displacing the subarachnoid space, non-enhancing, and displacing chronic vessels. The MRI-clinical correlation was telling—compression via the vermis for truncal ataxia, hemispheres for dysmetria and dysdiadochokinesia, peduncles for pursuing instability, and medulla for bulbar signs. Severe compression of the fourth ventricle and effacement of the cisterns defined surgical urgency and targets that day.

Surgery followed a simple principle: we relied on native craniospinal and cisternal planes to maintain safe brainstem corridors. Indeed, the lesion extending through the foramen of Magendie into the fourth ventricle, which occupied the cerebellomedullary cisterns, and conformed to the dorsal medulla and lower pons dictated the surgical approach; a midline suboccipital craniectomy with telovelar approach from the tonsillomedullary fissure was selected. The patient was positioned prone, with balanced flexion of the head to preserve venous drainage and align the axis of the foramen magnum. Reverse Trendelenburg positioning allowed assistance of gravity for the descent of the cerebellum after CSF release. After making a midline incision and splitting the muscle, we thinned the posterior rim of the foramen magnum and opened the dura in a Y fashion. The cerebellar tonsils were flattened and splayed; an incision through the arachnoid over the cisterna magna released plenty of CSF, decompressing them and providing relaxation of the cerebellum and opening the avascular tonsillomedullary fissure.

The tumor capsule, indicated by a pearly and laminated appearance, was extra-axial; it extended superior to the fourth ventricle, laterally along the cerebellomedullary cisterns, and inferior to the medullary tegmentum. The telovelotonsilar segment of PICA arched over the dome of the tumor and was identified early to preserve its perforators. Dorsal capsulotomy allowed for aspiration of keratin-like content with copious irrigation. Quadrant by quadrant internal decompression avoided asymmetrical collapse and created working space without retraction. Circumferential dissection progressed along arachnoid planes, preserving cranial nerves IX-XI, which retained their arachnoid sleeves. Superior adhesions were sectioned to free the choroid plexus to re-establish CSF continuity. At the floor of the fourth ventricle, where the capsule was securely adhered, I left behind thin avascular residue to avoid ependymal penetration and injury to the nucleus. PICA and branches were maintained, arachnoid membranes were left intact to avoid vasospasm, and bipolar coagulation was limited to avascular tissue.

Upon relieving adhesions, the brainstem regained convexity, and the tonsils their curvature, with rhythmic pulsations of CSF confirming restored flow. The foramen of Magendie and lateral recesses were patent, and the choroid plexus floated freely. The resection cavity was consistent with the original tumor bed without residual compression. Hemostasis was achieved without thermal alteration. We reconstructed the dura tension-free, approximated the musculature back to its original state, and closed the skin meticulously. Total intravenous anesthesia and continuous neurophysiological monitoring intraoperatively (brainstem auditory evoked potentials and cranial nerve IX–XII EMG) were employed to maintain brainstem and lower cranial nerve function. Prophylactic cefazolin (2 g IV) was administered upon induction and repeated 4 h after. Dexamethasone (8 mg IV bolus) was started for postoperative edema and weaned off over 48 h. Estimated blood loss was 180 mL with no transfusion. An external ventricular drain (EVD) was placed prior to the dural opening in the right frontal horn performed via a standard Kocher’s point entry (2.5 cm lateral to midline/1 cm anterior to coronal suture) under sterile technique. It was advanced freehand to a depth of approximately 5.5 cm until clear CSF was obtained. Controlled CSF drainage was started immediately after dural opening, and continued gradually throughout our maneuver to alleviate posterior fossa pressure, provide cerebellar relaxation, and decrease the chances of upward herniation during decompression. The EVD remained postoperatively for continuous intracranial pressure (ICP) monitoring and was clamped after 48 h once the ventricular size and CSF flow both remained stable on imaging. It was removed on postoperative day 3 without complication. Controlled release of CSF over the arachnoid incision with ventricular drain in place allowed for gradual decompression, and to avoid rapid decompression that may place the brainstem in tension. ICP remained physiologic throughout the surgery and postoperative period. Hemostasis was maintained by simple tamponade (without excess retraction), in addition to oxidized cellulose and fibrin sealant without bipolar coagulation near the ventricular floor. The hope is these measures limit secondary neurologic injury, maintain good visualization of the field, and enjoy neurologic recovery in the postoperative period.

After extubation in the neurosurgical ICU postoperatively, early recovery included trunk stability, improved gaze-holding nystagmus, and less dysmetria. There were no cranial nerve impairments noted, and swallowing remained safe. Postoperative CT (Figure 3) confirmed complete decompression and re-expansion of the fourth ventricle, patency the cerebellomedullary cistern, symmetric cerebellar hemispheres, and patent foramen of Magendie. There was no presence of hemorrhage, infarction or hydrocephalus, and the operative cavity duplicated the original tumor size, underscoring the painstaking extra-axial dissection while maintaining parenchymal margins.

An immediate postoperative MRI (Figure 4) was performed to assess the resection extent and surgical purpose. Multiplanar T1-weighted sequences showed a sharply demarcated postoperative cavity; the fourth ventricle had expanded, all cerebrospinal fluid paths were unobstructed, and there was no abnormal enhancement seen. The imaging findings served as direct visual proof for the intraoperative aims, namely, decompression of the brainstem, anatomical re-establishment of the cisterns, and maintenance of surrounding neurovascular structures. No perilesional edema and no hemorrhagic debris support the use of the “stay-at-anatomy” dissection approach. Near-total resection with a midline suboccipital telovelar approach accomplished complete decompression, anatomical restoration, and preservation of all appropriate neurovascular structures. Immediate postoperative MRI demonstrated no residual mass effect or abnormal enhancement, with no imaging assessment of keratinaceous content, despite the fact that there was microbiologically adherent capsular remainder on the ventricular floor that was purposefully left intact.

The early postoperative course revealed consistent, predictable, and gradual return of function, with subtle but significant daily neuro gains. The postoperative course was consistent and predictable return of function, but also daily neuro gains that were subtle but significant. After a few hours into her postoperative course, you could see she had an observable decrease in her preoperative cerebellar imbalance, with improved smoothness of limb movement and significantly less reliance on grosser-based stability. By the second day of postoperative recovery, her truncal stability improved enough for her to sit and stand, without the compensatory sway exhibited in her sitting and standing positions pre-surgery. Limb dysmetria was less prominent overall, and the rapid alternating movements that were previously hesitant and erratic were smooth in quality. Cranial nerve functions were intact as well, exhibiting intact oculomotor coordination, equal expression of face, stable swallowing, and phonation, suggesting that both cerebellar and brainstem pathways were preserved when dissected out.

Her mobilization was slow; however, by day four postoperative, she was ambulatory (i.e., independent), demonstrated dynamic balance with a better regard for spatial awareness. Her speech clarity overall remained normal, and there were no signs of bulbar compromise. An initial follow-up CT scan performed one week postoperatively (Figure 5) reaffirmed what was visually apparent at bedside: her fourth ventricle was entirely re-expanded; cerebellomedullary cistern remained patent; and posterior fossa structures remained in a reconstructed relationship. Radiologically, there were no signs of residual mass, hemorrhage, infarct, or new hydrocephalus. The converging relationship of anatomical restoration evident on imaging and gradual, quantifiable neuro recovery demonstrated the value of performing a dissection aligned to more native anatomical planes, with the maintained surrounding structures.

At the three-month postoperative assessment, this case sought to document not only the anatomical outcome, but the durability of functional restitution. The neurological examination demonstrated stable and complete resolution of the neurological deficits that initiated the presentation. The truncal instability that preoperatively presented as a wide-based gait and inability to perform tandem steps without loss of balance was fully resolved. The patient maintained midline posture without compensatory sway, including while standing with eyes closed, suggesting that the midline cerebellar control was restored. A three-month follow-up showed the KPS score was 90 and the mRS score was one. Testing of limb coordination revealed smooth and uninterrupted performance on finger-nose-finger and heel-knee-shin maneuvers with no terminal dysmetria or rebound phenomena. The rapidly alternating movements were both symmetric and fluid, with the previously noted dysdiadochokinesia on the ipsilateral side disappearing. The preoperative ocular pursuit instability and intermittent nystagmus could no longer be elicited, with horizontal and vertical gaze steady, full, and conjugate, and the initiation of saccades being instantaneous. Bulbar function had returned to normal with clear speech, coordinated swallowing, and no choking or nasal regurgitation while drinking liquids. There were no long tract signs, sensory asymmetries, or pathologic reflexes, and muscle tone was physiologic in all extremities. The patient also reported uninterrupted sleep, no further vertiginous episodes, and no headaches, indicating a sustained normalization of ICP dynamics. Daily living had been resumed fully, including occupational demand, without any limitations or accommodations.

A magnetic resonance image performed at this time (Figure 6) sought to provide an objective confirmation of surgical and clinical impressions. The axial T1-weighted post-contrast image (Figure 5A) demonstrated a sharply marginated, extra-axial postoperative cavity occupying the midline posterior fossa, with preservation of surrounding parenchymal architecture and no pathologic enhancement to suggest residual epidermoid tissue. There was no gliotic signal change or loss of regional volume, and the normal folial pattern of both cerebellar hemispheres was intact. The sagittal post-contrast T1-weighted image (Figure 5B) demonstrated a fourth ventricle in appropriate anatomical alignments, with unobstructed continuity between the aqueduct of Sylvius, the ventricular outlet foramina, and the cerebellomedullary cistern. The relationships between the cerebellar vermis, brainstem, and dorsal medulla displayed anatomical preservation, with no features suggestive of postoperative adhesive arachnoiditis or CSF flow obstruction.

The intended course for this case was to provide long-term decompression of the brainstem and restore cisternal anatomy while avoiding a secondary neurological compromise. At this moment, clinical and radiological sequelae indicated movement toward those ends. The absence of radiological recurrence, the preservation of surrounding neural structures, and the completeness of functional restitution all aligned with the goals outlined in advance of surgery. Ongoing structured clinical and radiological surveillance was planned with the recognition that even in cases demonstrating early and complete remission, continued watchfulness is an essential component of safely managing long-term outcomes.

A process of postoperative follow-up was planned to monitor recurrence, CSF flow abnormalities, and new neurologic deficits. The follow-up plan included clinical follow-up and MRI at 3 months (the patient should be functionally recovered and have no residual lesion), then at 12 months (a planned evaluation of long-term stability and no presence of adherent arachnoid changes), then yearly to assess for recurrence or delayed complication. The patient was informed of the early symptoms of CSF flow abnormality or increased ICP and encouraged to report any new symptoms as soon as they arise.

Patient perspective: The patient expressed that before surgery, her progressive imbalance and speech issues greatly impacted her ability to be independent and enjoy life. The patient was elated to regain balance and coordination within weeks of surgery, and she was pleased that she returned to her normal activities quickly, including resuming work expectations within 3 months. The patient underscored the value of the thorough preoperative counseling/education to explain the risks and recovery expectations.

## 3. Discussion

This case illustrates a rare presentation: dominant cerebellar dysarthria with severe truncal instability due to a large fourth-ventricular epidermoid—and highlights outcome-quantified, function-preserving telovelar microsurgery with a defined diffusion-weighted MRI follow-up protocol.

Surgical management of epidermoid cysts in the fourth ventricle is a unique thaw in posterior fossa neurosurgery, both because of the rarity of this lesion in this location, and because of the unique combination of its lethargic, insinuative growth pattern and the crowded eloquent anatomy of the brainstem–cerebellar complex [14]. In our case, the lesion had extended in a manner that was congruent with the classical cisternal and subarachnoid spread described in the published literature, with circumferential contact of the ventricular floor and cranial nerve root entry zones but not the parenchyma. The operative field validated that the lesion presented with a thin, avascular capsule that draped over the tela choroidea and inferior medullary velum, with focal tethering to branches of the PICA—a finding that is often cited as limiting radical clearance of the lesion. We therefore proceeded with dissection as a planned extra-axial separation as opposed to a traditional en bloc removal, a detail that echoed the increasingly popular surgical approach in modern series. Before surgery, this patient had a KPS of 60 and mRS of three, vestibular/balance deficits (Ataxia Rating Scale 34/56) and vestibulo-ocular deficits (including gaze-evoked nystagmus, smooth pursuit interruptions); cranial nerves were grade I bilaterally, except for subtle bilateral bulbar discoordination. By 7 days post-op, KPS was 80, mRS was two, ambulation was independent, and trunk sway decreased. By 3 months, KPS was 90, mRS was one, and the patient had re-established tandem gait, smooth pursuit, and saccadic initiation was normal, with cranial nerves at grade I. These objective outcomes align with more recent series showing clinically meaningful functional improvement after resection of posterior fossa epidermoids, and support the plane-based, function-conserving strategy being utilized [15,16].

Limitations and Future Considerations: This is a single case report with a relatively short-term follow-up period, so the conclusions that can be drawn from this study are limited and would need to be confirmed through larger, multicenter series. In addition, this case highlights the role of careful anatomical microsurgery and long-term imaging follow-up, but we may find that future technologies will increasingly rely on artificial intelligence as part of a clinical workflow. Recent work indicates that AI integrated into imaging diagnostics, surgical planning, decision support, and pathology can enhance diagnostic accuracy, improve operational plans, and customize care for each patient. If we combine AI technologies with advanced microsurgery techniques, we may improve outcomes for patients undergoing complex posterior fossa surgery and prolonged survivorship outcomes [17].

Extent of Resection and Prognostic Implications

The lobulated appearance of the lesion, encroachment along subarachnoid spaces, non-enhancement, and diffusion restriction provide an epidermoid signature that differentiates from arachnoid cysts (CSF signal without restriction) and enhancing neoplasms (i.e., meningioma, hemangioblastoma, or metastasis). DWI (and, when possible, 3D-CISS/FIESTA) offers a more specific diagnosis and potentially helps to plan surgery in the face of fourth-ventricular pathology [18].

Published series describing fourth-ventricular epidermoids describe the achievement of gross total resection (GTR) in 50–70% of cases, and the necessity of a subtotal resection (STR) in 30–50% because the capsule was adherent to important neurovascular structures [19]. Our surgical choice to leave small adherent remnants on the ventricular floor coincides with the general movement toward a philosophy that favors functional preservation over aggressive radical clearance when adhesions are made to the local neurovascular supply of lower cranial nerve nuclei. There is support for this method from a meta-analysis showing that there are permanent lower cranial nerve deficits in a minority of patients (up to 20%) who have undergone aggressive stripping of the upper cervical spine; typically, patients experience persistent dysphagia or dysphonia [16,20]. In our patient, the absence of a postoperative lower cranial nerve deficit suggests that preserving these adhesion planes allows for a much better quality of life and has no clinically significant risk of compromise to recurrence-free survival that exceeds a decade in the case of near-total excision. To put our surgical decision-making into a bigger context, we reviewed representative reports that have discussed fourth ventricle epidermoid cysts, the extent of resection, and functional outcomes. Although the literature is sparse and largely based on small or single-center experiences, these reports do provide useful context for understanding how anatomical limitations, operative approaches, and follow-up methods determined how much was removed and ultimately influenced the long-term outcome. We have summarized selected series and case reports into Table 1. Our intention is not to be exhaustive but to give a sense of trends and observations to help contextualize our case within the continuum of published experience.

Relative Surgical Approaches

In choosing a telovelar approach, our awareness of different outcomes in large comparative studies exposing patients to caudal, telovelar, and transvermian approaches gave us confidence that there was a reduction in the frequency and severity of truncal ataxia and cerebellar mutism in the telovelar cohort, albeit with little difference in access to the whole ventricular cavity in either cohort. In our patient, the relationship with the natural corridor of the cerebellomedullary fissure made us select a telovelar corridor and avoid cutting through the vermis, remembering that the vermis is essential for midline Ces cerebellar structures that are affective for coordinated movement and verbal expression, once challenged by the cerebellar mutism. While we were not faced with intraoperative assistance from the endoscope in this context, we did regard the values of using an endoscope to observe for any issues obstructing the foramina of Luschka and Magendie. In the literature we refer to, it has previously been noted that endoscopic optics have identified residual tumor in the recess, but not visible in the line of sight of a microscope. Critical appraisal of studies using endoscopic suggests there can be an advantage in increasing operative time and complexity unnecessarily, when an example of whether to use an endoscope should be an example may be a variable degree of indentations, irregularities, or multilobulation within a tumorous extension [24,25]. In the case presented here, there were no intraoperative endoscopic adjuncts used. The operative field had adequate visualization of both foramina of Magendie and Luschka without the requirement of angled endoscopic evaluation. Therefore, our discussion regarding endoscopic techniques is more context-based on evidence from other series rather than outlining their use in this case.

Role of Modern Intraoperative and Imaging Technologies

Scientific literature comparing intraoperative DWI has described it as being very sensitive and able to identify small residuals; however, given the slow growth and extraordinary rarity of these lesions, it usually results in no clinical utility for this extensive imaging process. This situation only reinforces our point in posterior fossa surgery, mentioning earlier, our anatomical expertise should go a long way beyond relying on technology [26].

Molecular and Histopathological Aspects

Our analysis of the histopathology was keratinising stratified squamous epithelium atop laminated keratin, cholesterol clefts, and some isolated macrophages (CD68 positive), all of which performed in line with established database criteria and their beginnings. The histopathological finding of focal thickening of the capsule corresponded very well to the intraoperative finding of a hard area of adhesion, which then leads to fascinating new literature that describes capsule integrity and overexpression of extracellular matrix proteins such as laminin, fibronectin, and integrins [26]. For example, maybe elucidating some or all of the reasons why an area of a capsule will not separate at all or sufficiently to allow a safe division.

In a translational manner, future work exploring either enzymatic or pharmacological means to disrupt adhesion-related molecules may lead to advances in safe radical resection. Especially regarding deep lesions such as ours [27].

Differential Diagnosis and Radiological Pitfalls

The preoperative MRI for the patient demonstrated the classical appearance of restricted diffusion allowed an easy and confident differentiation from arachnoid cysts that follow CSF signals and exhibit no DWI restriction. But the slight T1 heterogeneity did prompt a first thought that the etiology may not be normal CSF content, but rather, proteinaceous cystic tumors. Literature describes how hemorrhagic or proteinaceous epidermoids can mislead the differentiation, especially within the fourth ventricle, where cystic medulloblastomas or ependymomas can create an overlap of features [28]. In our instance, high resolution multiplanar sequences and careful consideration of the insinuative growth pattern provided key insight into directing the surgical plan.

Postoperative course, recovery and rehabilitation

The smooth and uncomplicated neurological recovery, early immobilization, and lack of new cranial nerve deficits are direct reflections of ablative surgery to the preserved arachnoid planes and also timely surgical intervention prior to irreparable and significant brainstem injury. Multiple reports in the literature demonstrate that delayed or persistent hydrocephalus can occur postoperatively in upwards of 10%. Consequently, we provided close clinical and imaging follow-up in the early postoperative months [29]. We maintained unobstructed CSF pathways at the foramen of Magendie, which most likely improved the stance of secondary ventriculomegaly.

Long-term follow-up and recurrence patterns

While our patient has had normal imaging with no regrowth at the early and intermediate follow-up time frames, their lienancy is not a lose win. The impressive documented potential for late recurrence which can frequently occur more than a decade later, incorporated with the residual capsule in which can be sealed with a residual keratin-loaded capsule, mandates lifetime MRI follow-up. Published follow-up algorithms generally site yearly MRI at years 1–5 with gradually lengthened intervals based on modifiable risk, etc., while a residual capsule, even microscopic or non-visible to the naked eye, can remain dormant for the period prior to developing increased keratin accumulation [30].

Global insight and health economics

This case illustrates a more global neurosurgery perspective on how access to MRI and posterior fossa microsurgical expertise has a significant impact on treatment. Even in healthcare environments without access to these resources, delays allow the tumor to grow larger than ideal, which contributes to preoperative deficits, risks of surgery, and postoperative complications [31]. Cost analysis shows earlier intervention results in more quality-adjusted life years (QALYs) and decreases long-term rehabilitation needs, which points to the further systemic value of implementing structured care pathways for rare lesions, like fourth ventricular epidermoids [32].

Our experience also supports several important principles relevant to neurosurgery practice. Firstly, a functional-preserving surgical philosophy—purposefully leaving a small adherent capsule attachment on the ventricular floor to avoid cranial nerve damage—is still able to decompress and achieve total neurologic recovery. Secondly, it is crucial to correlate surgical findings with exam and radiographic findings, as the nuances of an examination, including gaze-evoked nystagmus, truncal sway, and cerebellar dysarthria, correlated with anatomical compressions and the operative focus. Thirdly, intraoperative and histologic evidence of focal capsule enlargement reinforces the future direction of advances in the field of molecular adhesion biology that include laminin, fibronectin, and integrins, as well as the possible enzyme or pharmacological dissection approaches. Finally, supplementing anatomic detail with advanced modalities, such as AI-assisted imaging segmentations and lifelong MRI path-following programs, are ways that even one rare case contributes to the emergence of evidence-based advances in posterior fossa surgery.

## 4. Conclusions

Epidermoid cysts located within the fourth ventricle represent the most technically demanding area encountered in neurosurgery for posterior fossa pathology, not due to malignancy proclivity but due to the insydious and inspatial invasion of eloquent neurovascular territory. The case presented here demonstrates that the surgical importance lies not solely on the extent of lesion resection (though that may be significant), but also importantly, the neurologic and functional sequelae when the capsule adheres to the brainstem or lower cranial nerve nuclei.

Our experience suggests that a telovelar, vermis-sparing approach with careful arachnoidal dissection, albeit difficult, provides adequate decompression and an acceptable long-term potential outcome with a low probability of new deficits. In this instance, and on conceptual grounds and “maximal safe resections,” the rationale for not leaving the capsule undisrupted was motivated more by the desire to maximize patient welfare, rather than to achieve a histologically complete resection.

In all fairness, we acknowledge that the consequences of a single case from a relatively short-term follow-up are generally anecdotal. Our early success needs to be confirmed long-term because recurrences have been identified even years to decades postoperatively. To aid that process, we have developed a follow-up protocol employing diffusion-weighted MRI beginning at twelve months’ follow-up and then annual follow-ups for five years, and then bi-annual follow-ups. Only time will clarify the long-term treatment success and efficacy. Future work regarding imaging protocols and better understanding the underlying molecular pathophysiology related to capsule adherence will possibly assure more complete and safer resection in the future. Until that time, this case adds minimally to the literature but endorses and promotes a distillate anatomical position and promotes a dogged surgical philosophy: to carry out resection with consideration to neurovascular safety, to perform the least aggressive procedures with the best functional outcomes, and long-term follow-up.

## Figures and Tables

**Figure 1 diagnostics-15-02600-f001:**
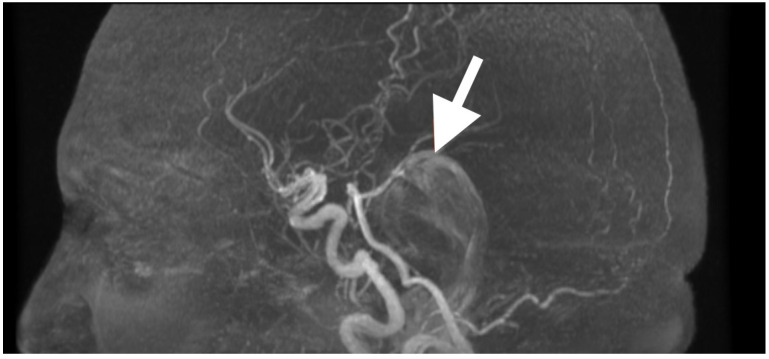
Preoperative magnetic resonance angiography, sagittal reconstruction, demonstrating a large extra-axial mass in the left posterior fossa centered in the cerebellar hemisphere with medial extension toward the vermis. The lesion produces smooth elongation and displacement of adjacent posterior fossa arteries, most prominently the posterior inferior cerebellar artery (PICA, white arrow), which is draped over the superior–posterior surface of the mass. Vessel continuity and caliber are preserved, indicating chronic adaptive displacement rather than acute encasement or invasion. The absence of abnormal vascular blush or arteriovenous shunting supports the impression of an avascular lesion, consistent with the imaging profile of an epidermoid tumor.

**Figure 2 diagnostics-15-02600-f002:**
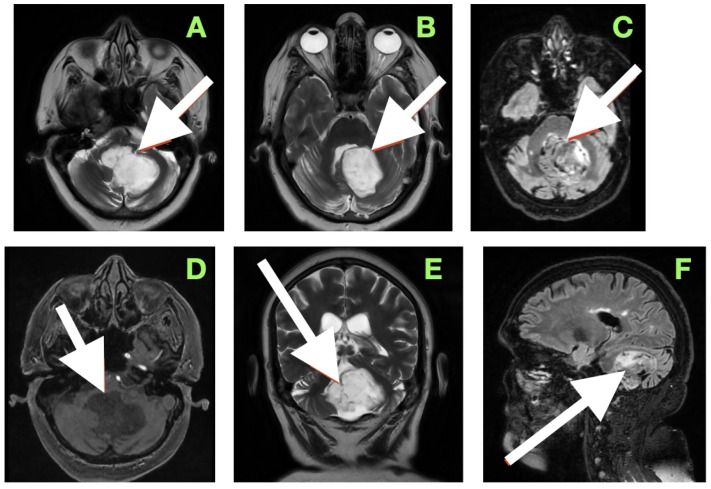
Preoperative magnetic resonance imaging of the posterior fossa. (**A**,**B**) Axial T2-weighted images reveal a large lobulated hyperintense lesion occupying the left cerebellar hemisphere, extending medially into the vermis and inferiorly into the cisterna magna, with partial effacement and anterior displacement of the fourth ventricle. The lesion insinuates between cerebellar folia without a discrete capsule, a feature characteristic of epidermoid tumors. (**C**) Susceptibility-weighted imaging shows punctate hypointense foci within the lesion, likely representing calcific or hemosiderin deposits from chronic contact with adjacent neurovascular structures. (**D**) Axial post-contrast T1-weighted image demonstrates absence of enhancement, distinguishing the lesion from hypervascular posterior fossa neoplasms. (**E**) Coronal T2-weighted image depicts inferior extension toward the foramen of Luschka and compression of the cerebellar peduncles, explaining the patient’s gait instability, gaze-evoked nystagmus, and dysdiadochokinesia. (**F**) Sagittal FLAIR sequence confirms anterior displacement and flattening of the dorsal medulla with partial effacement of the prepontine cistern, correlating with the subtle bulbar incoordination observed on examination.

**Figure 3 diagnostics-15-02600-f003:**
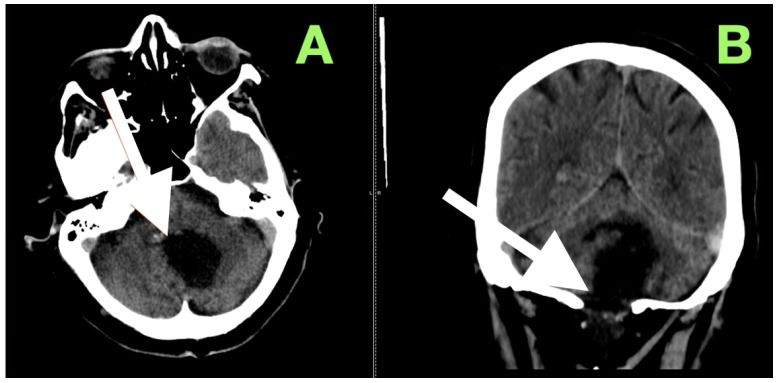
Immediate postoperative non-contrast CT scan demonstrating complete removal of the fourth-ventricular epidermoid tumor and restoration of posterior fossa anatomy. (**A**) Axial view shows a well-defined postoperative cavity in the midline vermian and fourth-ventricular region, with full re-expansion of the fourth ventricle and reappearance of the cerebellomedullary cistern. The brainstem contour is normalized, and no residual mass effect or obstructive hydrocephalus is present. (**B**) Coronal reconstruction confirms symmetrical cerebellar hemispheres, midline restoration of the fourth ventricle, and patent foramen of Magendie, with no acute hemorrhage or postoperative edema. The surgical cavity follows the anatomical boundaries of the preoperative lesion, reflecting precise extra-axial microsurgical dissection and preservation of surrounding neural tissue.

**Figure 4 diagnostics-15-02600-f004:**
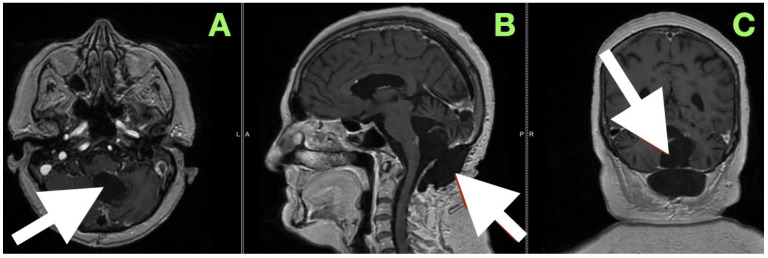
Immediate postoperative MRI. (**A**) Axial, (**B**) sagittal, and (**C**) coronal T1-weighted images show a postoperative cavity with complete re-expansion of the fourth ventricle and patent CSF pathways (white arrows). No abnormal enhancement or diffusion-restricted focus is present, confirming gross total resection and restoration of posterior fossa anatomy.

**Figure 5 diagnostics-15-02600-f005:**
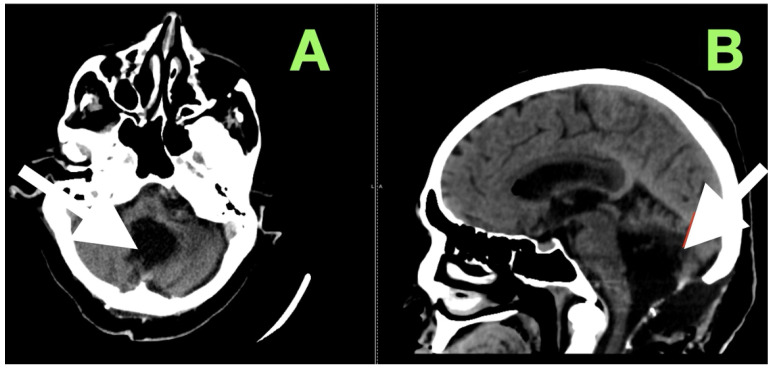
One week postoperative non-contrast CT scan demonstrating sustained decompression of the posterior fossa and stable restoration of normal anatomical relationships. (**A**) Axial view shows a well-defined postoperative cavity in the region previously occupied by the epidermoid tumor, with persistent re-expansion of the fourth ventricle and preservation of the brainstem contour. The cerebellomedullary cistern remains patent, and there is no evidence of recurrent mass effect, hemorrhage, or hydrocephalus. (**B**) Sagittal reconstruction confirms midline restoration, stable ventricular configuration, and durable decompression of the aqueduct–fourth ventricle complex, with preservation of surrounding cerebellar and brainstem tissue. The imaging appearance is consistent with complete extra-axial removal and continued functional stability.

**Figure 6 diagnostics-15-02600-f006:**
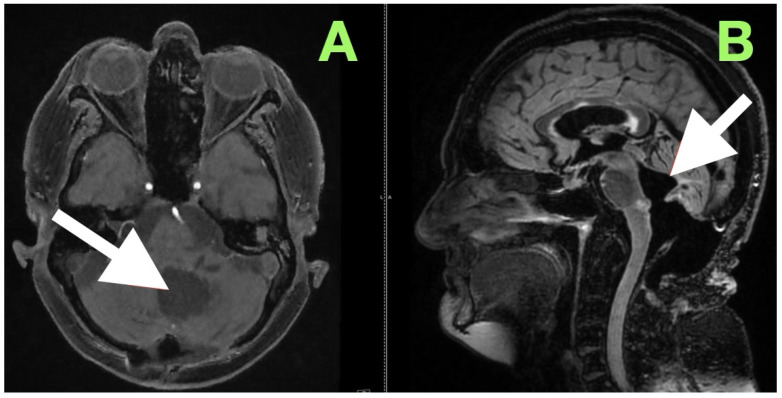
Three-month postoperative contrast-enhanced MRI documenting stable anatomical restoration and absence of residual lesion. (**A**) Axial post-contrast T1-weighted image shows a sharply delineated postoperative cavity in the midline posterior fossa, with preserved cerebellar hemisphere contour and absence of abnormal enhancement. (**B**) Sagittal post-contrast T1-weighted image demonstrates a fully re-expanded fourth ventricle, patent CSF pathways, and stable alignment of posterior fossa structures, without recurrence or postoperative complication.

**Table 1 diagnostics-15-02600-t001:** Selected published series and case reports focusing specifically on fourth-ventricular epidermoid cysts, highlighting surgical approaches, extent of resection, and postoperative outcomes.

Study/Year	Anatomical Focus	Key Findings	Reported Outcomes & Follow-Up	Reference
Kumar et al., 2021	Cases of fourth-ventricular epidermoid cysts (systematic review + case series)	GTR achieved in ~61%; STR required in ~39% due to capsule adherence	Median recurrence ~7 years; lifelong follow-up advised	[14]
Hanaei et al., 2024	Fourth ventricular tumors (mixed histologies)	Telovelar approach reduced cerebellar mutism and truncal ataxia vs. transvermian	GTR ~68%, STR ~32%; Permanent cranial nerve deficits < 10%	[21]
Pettersson et al., 2023	Multicenter cohort, fourth ventricle tumors	Surgical approach strongly influenced postoperative morbidity	Permanent cranial nerve deficits < 10% with telovelar vs. ~20% with transvermian	[22]
Trivedi et al., 2022	Giant posterior fossa epidermoids, fourth ventricle subset	GTR linked to lowest recurrence; STR indicated when capsule adherent to floor	Permanent lower cranial nerve deficits ~15% with aggressive resections	[23]
Hasegawa et al., 2021	Intracranial epidermoids, including 9 in the fourth ventricle	Extent of resection correlates directly with recurrence-free survival	Recurrence ~10% at 10 years (GTR) vs. ~35% (STR)	[15]

## Data Availability

The original contributions presented in this study are included in the article. Further inquiries can be directed to the corresponding authors.

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
