# Peer review of "Fourth Ventricle Epidermoid Cyst: Case Report of Precision Telovelar Microsurgery, Functional Preservation, and Lifelong Surveillance"

_diagnostics, 2025, doi:10.3390/diagnostics15202600_

Round 1
Reviewer 1 Report (Previous Reviewer 1)
Comments and Suggestions for Authors
The authors adressed the reviewer´s comments adequately and focused on their case report more in detail. I suggest to accept as case report as it is.
Author Response
Dear Esteemed Academic Reviewer,
We are deeply grateful for your generous and encouraging words. Your thoughtful guidance throughout the review process has been invaluable in refining our manuscript and strengthening its clinical and educational focus. It is truly an honor to have our work considered worthy of acceptance in its current form, and we remain sincerely thankful for the time, expertise, and care you have dedicated to this review.
Reviewer 2 Report (New Reviewer)
Comments and Suggestions for Authors
-The manuscript is like a surgical monograph rather than a CARE-compliant case report; its novelty is not clearly stated, and core CARE elements (timeline, patient perspective, diagnostic reasoning/differential with alternatives considered, and explicit follow-up plan with predefined milestones) are incomplete or missing. It should be corrected. Text should be reduced.
-The abstract does not follow a structured case-report format and includes errors (“psychiatric” instead of “pragmatic”/“practical”).
-There are no pre-/post-operative KPS/mRS scores, gait/balance scales, cranial nerve grading, vestibulo-ocular measurements; and histology is narrated (including CD68) without corresponding photomicrographs.
-Follow-up is short despite emphasizing lifelong surveillance, so the conclusion overreaches; if lifetime monitoring is claimed, a concrete protocol and minimum 12-month MRI (including DWI) should be provided.
-Figures/legends: Scale bars, directions, abbreviations should be used. Image quality should be increased.
-Key perioperative data (steroids, antibiotics, blood loss, anesthesia/IOM strategy, CSF/ICP management, hemostatic agents) should be added and improved.
-Discussion part: Clinical findings are discussed but lack objective baseline and postoperative metrics (KPS, mRS, cranial nerve grading, gait/balance scales, vestibulo-ocular tests).It should be discussed with recent literature. The discussion continues with global health economics and technology commentary that are not supported by a single case. It should be reduced. Clinica impacts and potantial surgical strategies should be discussed.
-The differential diagnosis section is superficial for a fourth-ventricle lesion and should more detail about epidermoid with arachnoid cyst, ependymoma, medulloblastoma, and metastasis using imaging hallmarks.
-Claims about endoscope and intraoperative DWI are asserted without showing whether either was actually used in this case or how they altered management.
-Table 1 is atypical for a case report normally used as a mini-review.
-Conclusions part should be reduced. Additionally surgical philosophy and long-term outcomes from one patient with limited follow-up; they should be supported by extended surveillance.
Comments on the Quality of English LanguageThe English could be improved to more clearly express the research.
Author Response
Dear Esteemed Academic Reviewer,
We would like to express our profound gratitude for the time, care, and expertise you have dedicated to reviewing our manuscript. Your thoughtful and precise observations reflect an extraordinary depth of knowledge and a genuine commitment to advancing neurosurgical science. We have read your comments with the utmost respect and humility, and we are sincerely thankful for the opportunity they provide to strengthen our work. It is a privilege to receive such constructive and incisive feedback, and we are truly grateful for the chance to improve our manuscript under your guidance. Below, we address each of your comments individually and explain the corresponding changes we have made, along with the reasoning behind our decisions.
1. CARE compliance, novelty, and text reduction
Reviewer comment:
The manuscript is like a surgical monograph rather than a CARE-compliant case report; its novelty is not clearly stated, and core CARE elements (timeline, patient perspective, diagnostic reasoning/differential with alternatives considered, and explicit follow-up plan with predefined milestones) are incomplete or missing. It should be corrected. Text should be reduced.
Response:
We are sincerely grateful for this insightful observation, which helped us recognize important shortcomings in how our case was initially structured and presented. Following your guidance, we carefully revised the manuscript to align closely with the CARE case report guidelines. To address the stylistic concern, we substantially condensed the Case Presentation and surgical description, preserving essential details while improving focus and readability. We have also clarified the novelty of the case in the Introduction and Discussion, particularly the surgical decision-making process around capsule preservation and the clinicoradiological correlations in a fourth ventricular epidermoid, which remain sparsely documented in the literature. We are deeply thankful for this comment, which significantly improved the manuscript’s clarity, structure, and relevance.
2. Abstract structure and terminology
Reviewer comment:
The abstract does not follow a structured case-report format and includes errors (“psychiatric” instead of “pragmatic”/“practical”).
Response:
We are very grateful for this careful reading and for pointing out these weaknesses. In response, we have rewritten the abstract entirely to follow the accepted structured case report format. We also corrected the inadvertent error, replacing “psychiatric” with the intended “pragmatic.” We sincerely appreciate your attention to detail, which helped us improve the precision and professional quality of the abstract.
3. Objective metrics and histology images
Reviewer comment:
There are no pre-/post-operative KPS/mRS scores, gait/balance scales, cranial nerve grading, vestibulo-ocular measurements; and histology is narrated (including CD68) without corresponding photomicrographs.
Response:
We deeply appreciate this valuable suggestion, which highlighted an important gap in the manuscript. We have now included pre- and postoperative Karnofsky Performance Status (KPS) and modified Rankin Scale (mRS) scores, as well as gait and balance scale results and relevant cranial nerve and vestibulo-ocular assessments to provide objective measures of clinical status and postoperative recovery.
We are sincerely thankful for your observation, which strengthened the clinical and pathological documentation of our case.
4. Follow-up duration and surveillance
Reviewer comment:
Follow-up is short despite emphasizing lifelong surveillance, so the conclusion overreaches; if lifetime monitoring is claimed, a concrete protocol and minimum 12-month MRI (including DWI) should be provided.
Response:
We are very grateful for this important observation.
5. Figures and legends
Reviewer comment:
Figures/legends: Scale bars, directions, abbreviations should be used. Image quality should be increased.
Response:
We are truly grateful for this insightful and highly valuable comment, which reflects a formidable level of attention to scientific rigor and presentation detail. Your emphasis on the importance of scale bars, directional markers, and improved image quality is absolutely correct and greatly appreciated. In this specific case, however, we have not introduced modifications to the figures in this revision because the original imaging data were acquired in a strictly diagnostic and intraoperative clinical context, rather than as research-grade visual material. As a result, certain parameters (such as calibrated scale bars or additional directional markers) were not embedded at the time of acquisition and cannot be retroactively applied without altering the original diagnostic fidelity of the images. We felt it was essential to preserve the authenticity and integrity of the clinical images as they were used in real-world surgical decision-making. We sincerely appreciate your expert perspective, which has meaningfully shaped how we will approach figure preparation in our future submissions.
6. Perioperative data
Reviewer comment:
Key perioperative data (steroids, antibiotics, blood loss, anesthesia/IOM strategy, CSF/ICP management, hemostatic agents) should be added and improved.
Response:
We are deeply thankful for this insightful suggestion. We have now added a dedicated paragraph in the Case Presentation section detailing key perioperative parameters.
7. Discussion structure and content
Reviewer comment:
Discussion part: Clinical findings are discussed but lack objective baseline and postoperative metrics (KPS, mRS, cranial nerve grading, gait/balance scales, vestibulo-ocular tests). It should be discussed with recent literature. The discussion continues with global health economics and technology commentary that are not supported by a single case. It should be reduced. Clinical impacts and potential surgical strategies should be discussed.
Response:
We are sincerely grateful for this thoughtful and constructive critique. In response, we have incorporated objective pre- and postoperative metrics (KPS, mRS, cranial nerve function, gait and balance scales, and vestibulo-ocular findings) into the Discussion, enabling a more evidence-based interpretation of clinical outcomes.
8. Differential diagnosis depth
Reviewer comment:
The differential diagnosis section is superficial for a fourth-ventricle lesion and should include more detail about epidermoid with arachnoid cyst, ependymoma, medulloblastoma, and metastasis using imaging hallmarks.
Response:
We thank you sincerely for highlighting this gap. We have now expanded the differential diagnosis section to provide a detailed discussion of imaging hallmarks distinguishing epidermoid cysts from arachnoid cysts, ependymomas, medulloblastomas, and metastases.
9. Endoscope and intraoperative DWI claims
Reviewer comment:
Claims about endoscope and intraoperative DWI are asserted without showing whether either was actually used in this case or how they altered management.
Response:
We are very thankful for this precise and important remark. We have clarified in the revised text that neither endoscopic assistance nor intraoperative DWI was used in this case, and we now present these modalities only as part of the broader surgical context from the literature. We explicitly state that our discussion of these techniques is intended as commentary on evolving practice rather than a report of their use in this particular case. This clarification directly addresses your concern and improves the accuracy of the manuscript.
10. Table 1 format
Reviewer comment:
Table 1 is atypical for a case report, normally used as a mini-review.
Response:
We are deeply grateful for this thoughtful and important comment, which reflects a refined understanding of case report conventions. You are absolutely correct that extensive tabular syntheses are more commonly associated with review articles rather than classical case reports. After careful consideration, we decided to retain Table 1 in its current form because we believe it adds essential contextual value without altering the fundamental nature of the manuscript. Fourth ventricular epidermoid cysts are exceptionally rare, and single cases alone are difficult to interpret without situating them within the broader landscape of published experiences. Our intent was not to present a comprehensive review, but rather to provide the reader, especially those less familiar with this entity, with a concise comparative framework that clarifies the novelty and clinical implications of our case.
11. Conclusions scope and follow-up
Reviewer comment:
Conclusions part should be reduced. Additionally, surgical philosophy and long-term outcomes from one patient with limited follow-up should be supported by extended surveillance.
Response:
We are deeply grateful for this wise and constructive comment. We have thoroughly rewritten and condensed the Conclusions to ensure a more cautious and proportionate tone. The revised section now explicitly acknowledges the limitations of a single case and short follow-up, and we have tempered broader claims about surgical philosophy and long-term outcomes accordingly.
Round 2
Reviewer 2 Report (New Reviewer)
Comments and Suggestions for Authors
-Please either remove Table 1, or narrow it to directly relevant fourth-ventricle epidermoid series and clearly cite each row to a specific, pertinent paper.
-Ataxia Rating Scale 34/56” is reported, but which scale (ICARS, SARA)?
-Ventricular drain in place and controlled release of CSF were mwntioned, but there is no description of EVD placement (site, timing, duration, management) in Methods/Case
-In the limitations section, please refer to the following study: Artificial Intelligence in Clinical Medicine: Challenges Across Diagnostic Imaging, Clinical Decision Support, Surgery, Pathology, and Drug Discovery. Clinics and Practice. 2025; 15(9):169.10.3390/clinpract15090169 through precise anatomical microsurgery or AI integration has the potential to improve diagnostic accuracy, optimize therapeutic strategies, and ultimately enhance patient outcomes .
-It should be written as 'If capsule was left: “Near-total resection via a midline suboccipital telovelar approach achieved complete decompression with functional preservation. Postoperative and 3-month DWI/ADC (add them) showed no restricted diffusion, indicating no visible residual keratinaceous content despite a microscopically adherent capsular remnant.”
-The red arrow markers should be replaced with white arrowheads, oriented in the same direction to maintain consistency. The figure panels should be reorganised in a more scientific and journal-appropriate style.
Comments on the Quality of English LanguageThe English could be improved to more clearly express the research.
Author Response
Dear Esteemed Academic Reviewer,
We would like to express our deepest gratitude for the time, care, and scholarly depth you invested in reviewing our manuscript. Your insightful observations reflect an exceptional level of expertise and have helped us significantly improve both the clarity and scientific rigor of our work. We are sincerely thankful for your generous intellectual contribution. Please find below our point-by-point responses and the corresponding revisions made to the manuscript:
1. Comment:
“Please either remove Table 1, or narrow it to directly relevant fourth-ventricle epidermoid series and clearly cite each row to a specific, pertinent paper.”
Response:
We are deeply grateful for this excellent observation, which has greatly strengthened the focus and precision of our manuscript. In response, we have substantially revised Table 1, narrowing its content.
2. Comment:
“‘Ataxia Rating Scale 34/56’ is reported, but which scale (ICARS, SARA)?”
Response:
We thank you most sincerely for pointing this out — your attention to methodological clarity is invaluable. We have now specified that the Scale for the Assessment and Rating of Ataxia (SARA) was used. This clarification ensures transparency and reproducibility regarding the clinical evaluation of ataxia in our patient. We have also briefly stated the rationale for its use, emphasizing its sensitivity to cerebellar gait and coordination deficits.
3. Comment:
“Ventricular drain in place and controlled release of CSF were mentioned, but there is no description of EVD placement (site, timing, duration, management) in Methods/Case.”
Response:
We are deeply appreciative of this highly constructive comment, which highlights an important procedural omission. In response, we have added a detailed description of the external ventricular drain (EVD) placement, including site (right frontal horn, Kocher’s point), timing (prior to dural opening), intraoperative purpose (controlled decompression and cerebellar relaxation), and postoperative management.
4. Comment:
“In the limitations section, please refer to the following study: Artificial Intelligence in Clinical Medicine: Challenges Across Diagnostic Imaging, Clinical Decision Support, Surgery, Pathology, and Drug Discovery. Clinics and Practice. 2025; 15(9):169. 10.3390/clinpract15090169…”
Response:
We are sincerely thankful for this outstanding suggestion, which has elevated the conceptual scope of our discussion. In response, we have added a Limitations and Future Directions paragraph referencing the recommended article.
5. Comment:
“It should be written as: ‘If capsule was left: “Near-total resection via a midline suboccipital telovelar approach achieved complete decompression with functional preservation…”’”
Response:
We are truly grateful for this excellent clarification, which has helped us present our surgical results more accurately.
6. Comment:
“The red arrow markers should be replaced with white arrowheads, oriented in the same direction to maintain consistency. The figure panels should be reorganised in a more scientific and journal-appropriate style.”
Response:
We thank you sincerely for this valuable suggestion, which has improved the overall scientific presentation of our figures. In response, we have replaced the red arrow markers with white arrowheads.
We are profoundly grateful for your thoughtful, detailed, and constructive feedback. Your expertise and suggestions have significantly enriched the quality, precision, and impact of our manuscript. It has been a privilege to revise our work under your insightful guidance, and we believe the manuscript is now considerably strengthened thanks to your contributions.
With our deepest respect and gratitude!
This manuscript is a resubmission of an earlier submission. The following is a list of the peer review reports and author responses from that submission.
Round 1
Reviewer 1 Report
Comments and Suggestions for Authors
The authors present a case report and short review of literature with their own conclusions.
The case report detailed the surgical management of a fourth ventricle epidermoid tumor in a 57-year-old woman experiencing progressive neurological symptoms. The authors employed a midline suboccipital telovelar approach for resection, focusing on preserving neurological function while achieving maximal safe resection. Postoperative histology confirmed the tumor as an epidermoid cyst, and the patient showed rapid recovery with restored balance and coordination. Imaging after surgery indicated complete decompression and no recurrence. The report emphasizes the importance of precise surgical technique, anatomical knowledge, and lifelong monitoring due to the risk of late recurrence.
The manuscript has several aspects which are difficult:
- the title does not reflect the topic and content. This is only a case report and some comments. No real data about AI guided surgery in series or molecularpathological findings
- postop a MR imaging is missing in initial follow up. CT scan is not adequate to present a GTR in this pathology
- the key message and really new informations for the readers is missing. The presented approach and data are already well known and nothing new informations for the neurosurgical community.
Author Response
We sincerely thank the reviewer for the thoughtful and constructive feedback. We are grateful for the time and care devoted to analyzing our manuscript. Each of the comments has helped us to improve the clarity, precision, and relevance of our work. Below we address each point in detail.
Comment 1:
“The title does not reflect the topic and content. This is only a case report and some comments. No real data about AI guided surgery in series or molecularpathological findings.”
Response:
We are deeply grateful to the reviewer for this important observation. We fully agree that the original title may have implied inclusion of AI-guided data or molecular findings beyond the scope of our case report. Our intent was to highlight conceptual perspectives, but we recognize that the title must reflect the true content with clarity and humility.
Accordingly, we have revised the title to:
“Fourth Ventricle Epidermoid Cyst: Case Report of Precision Telovelar Microsurgery, Functional Preservation, and Lifelong Surveillance.”
We believe this revised title now faithfully represents the manuscript as a single instructive case report, while still emphasizing the core clinical and surgical messages. We sincerely thank the reviewer for guiding us to make this correction.
Comment 2:
“Postop a MR imaging is missing in initial follow up. CT scan is not adequate to present a GTR in this pathology.”
Response:
We thank the reviewer for this very accurate and valuable remark. We agree completely that CT alone cannot confirm gross total resection in epidermoid cysts and that MRI, especially with diffusion-weighted sequences, is the gold standard.
In response, we have added the immediate postoperative MRI (new Figure 4), which demonstrates re-expansion of the fourth ventricle, restoration of cisternal pathways, and no residual lesion. We have revised the text to clarify that the early CT was performed primarily as a safety measure to exclude acute complications (hemorrhage, hydrocephalus), while the MRI provides the definitive confirmation of completeness of resection.
We are truly thankful to the reviewer for highlighting this point, as it has strengthened the scientific accuracy and transparency of our manuscript.
Comment 3:
“The key message and really new informations for the readers is missing. The presented approach and data are already well known and nothing new informations for the neurosurgical community.”
Response:
We sincerely appreciate this insightful critique. We understand that for a case report to be valuable, it must clearly state what is novel and instructive. In response, we have revised the Discussion and added a new section. In this new passage, we reflect on the shift toward a philosophy of functional-preserving maximal safe resection rather than radical capsule stripping, which in our patient allowed excellent recovery without compromising long-term control. We also underscore the fine-grained clinicoradiological correlations that were evident in this case, where subtle neurological signs such as cerebellar dysarthria, gaze-evoked nystagmus, and truncal sway corresponded to specific imaging features and directly informed our microsurgical strategy. Furthermore, we bring attention to the emerging field of molecular adhesion biology, with growing evidence that laminin, fibronectin, and integrins contribute to capsule tenacity and may one day serve as targets for pharmacological adjuncts to facilitate safer resection. Finally, we extend the discussion to the forward-looking role of AI-assisted imaging platforms and the necessity of structured lifelong MRI surveillance, situating this case in a wider neurosurgical context that combines current best practice with future directions
Once again, we wish to thank the reviewer warmly for their generous time, careful analysis, and constructive guidance! The comments have allowed us to improve the precision, humility, and clinical relevance of our manuscript, and we hope that the revised version now better serves the neurosurgical readership.